# Immunomodulatory Effects of TGF-β Family Signaling within Intestinal Epithelial Cells and Carcinomas

**Paula Marincola Smith [1],\*** , **Anna L. Means [1,2] and R. Daniel Beauchamp [1,2]**

1   Department of Surgery, Section of Surgical Sciences, Vanderbilt University Medical Center, Nashville, TN 37232, USA; anna.means@vumc.org (A.L.M.); daniel.beauchamp@vumc.org (R.D.B.)
2   Department of Cell & Developmental Biology, Vanderbilt University, Nashville, TN 37232, USA
\*   Correspondence: paula.m.smith@vumc.org; Tel.: +1-615-343-2735

**Abstract:** TGF-β superfamily signaling is responsible for many critical cellular functions including control of cell growth, cell proliferation, cell differentiation, and apoptosis. TGF-β appears to be critical in gastrulation, embryonic development, and morphogenesis, and it retains pleiotropic roles in many adult tissues and cell types in a highly context-dependent manner. While TGF-β signaling within leukocytes is known to have an immunosuppressive role, its immunomodulatory effects within epithelial cells and epithelial cancers is less well understood. Recent data has emerged that suggests TGF-β pathway signaling within epithelial cells may directly modulate pro-inflammatory chemokine/cytokine production and resultant leukocyte recruitment. This immunomodulation by epithelial TGF-β pathway signaling may directly impact tumorigenesis and tumor progression through modulation of the epithelial microenvironment, although causal pathways responsible for such an observation remain incompletely investigated. This review presents the published literature as it relates to the immunomodulatory effects of TGF-β family signaling within intestinal epithelial cells and carcinomas.

**Keywords:** TGF-beta; transforming growth factor beta; colon cancer; inflammatory bowel disease; IBD

## 1. TGF-β Family Signaling

The TGF-β superfamily is comprised of over thirty distinct, secreted cytokines (including TGF-βs, Bone Morphogenic Proteins (BMPs), Nodal, and Activin) [1] that perform many cellular functions including control of cell growth, cell proliferation, cell differentiation, and apoptosis [1–4]. TGF-β family signaling appears to be critical for gastrulation, embryonic development, and morphogenesis, and it has pleiotropic roles in many adult tissues and cell types. The impact of TGF-β family pathway signaling is highly cell type- and context-dependent.

TGF-β ligands bind to a family of TGF-β cell surface receptors, which are present on most cell types in the body, and include TGF-βRII, TGF-βRI, BMPR2, BMPR1A/1B, ACVR2A/2B, and ACVR1A/1B [1]. In the case of the TGF-β receptors, TGF-βRII is constitutively active, and upon ligand binding, the type II receptors activate the type I receptors via transphosphorylation and form a hetero-tetrameric complex composed of two TGF-βRIIs and two TGF-βRIs [4]. Upon TGF-βR activation and complex formation, downstream signaling is perpetuated via two major routes: SMAD-dependent (canonical) and SMAD-independent (non-canonical) signaling [5]. The canonical signaling pathway is the more well-characterized pathway, whereas the non-canonical pathway is less well understood, and its biological relevance remains less clear. In canonical signaling, TGF-βR activation leads to phosphorylation of the receptor-regulated SMADs (R-SMADs), which include SMADs 2 and

3 in the case of TGF-βRs and SMADs 1, 5, and 9 in the case of BMP Receptors (BMPRs). After phosphorylation/activation, the R-SMADs associate with the common partner SMAD (co-SMAD), SMAD4, before translocation to the nucleus [3]. Once in the nucleus, the SMAD complexes bind directly to DNA via their MH1 domain and regulate transcription via their MH2 domain [3,4,6]. The inhibitory SMADs (I-SMADs), SMADs 6 and 7, are induced by canonical TGF-β pathway signaling and function to block R-SMAD phosphorylation and R-SMAD/SMAD4 complex formation, thus negatively regulating TGF-β pathway signaling [5,7]. Of note, there is some evidence to suggest R-SMADs may function independently of SMAD4 in some circumstances [8], although these pathways remain incompletely investigated.

Through canonical TGF-β family member signaling, SMAD complexes interact with a wide variety of distinct DNA binding sites and target genes. Importantly, once in the nucleus, SMAD complexes require the cooperation of cofactors (coactivators and corepressors) to successfully bind DNA and regulate transcriptional programs. The transcriptional program induced by the TGF-β family signaling pathway via SMAD proteins is, thus, highly cell type- and context-specific, as the presence or absence of various cofactors can have a dramatic impact on SMAD-target gene interactions [6]. Recent research suggests that SMAD complexes determine their target sites along with other DNA-binding cofactors by two distinct mechanisms. First, cell type- or lineage-specific transcriptional cofactors open chromatin at specific SMAD binding elements (SBEs), making certain that SBEs are accessible to nuclear SMAD complexes. Second, DNA-binding cofactors, induced and activated in a context-dependent manner, can directly strengthen the interaction between SMAD complexes and DNA. The result of this cofactor dependence is that the downstream effects of TGF-β superfamily canonical signaling may differ based on the cell type and context in which it is delivered, thus causing significant heterogeneity in TGF-β superfamily signaling responses between different tissues and within tissues at different stages of development or differentiation. This also means that TGF-β response data from cell culture experiments should be interpreted with caution.

The non-canonical TGF-β signaling pathways are less well-characterized, but may play important roles in regulating many TGF-β pathway functions through three distinct mechanisms: non-SMAD signaling pathways that directly modify SMAD function, non-SMAD proteins whose function is directly modulated by SMADs and which transmit signals to other pathways, and non-SMAD proteins that directly interact with or become phosphorylated by TGF-β receptors and do not necessary affect SMAD function. Some signaling molecules that have been implicated in non-SMAD TGF-β signaling include various elements of the Mitogen-activated protein (MAP) kinase pathway (including Erk and JNK/p38 activation) [9–12], Rho-like GTPase signaling pathway [13,14], and phosphatidylinositol-3-kinase/AKT pathway [15–18]. These collective SMAD-independent pathways appear to affect target cells by promoting apoptosis and cellular differentiation, impinging on cell proliferation, contributing to epithelial to mesenchymal transition (EMT), and modulating matrix regulation [19]. These non-canonical TGF-β signaling activities, especially those that are involved with cytoskeletal remodeling and EMT, are of particular importance in understanding TGF-β's duality of function between tumor prevention and tumor promotion (described in more detail in the following section: TGF-β pathway dysregulation in cancer). A complete review of the SMAD-independent pathways is beyond the scope of this paper, and this topic has been previously reviewed by Moustakas and colleagues [5] as well as Zhang [19]. Nonetheless, it is important to acknowledge that the SMAD-independent pathways likely impinge on the highly context-specific responses to TGF-β signaling, and that these pathways are deserving of further investigation.

## 2. TGF-β Pathway Dysregulation in Cancer

Various components of the TGF-β signaling pathway are frequently reported lost or dysregulated in multiple types of cancer. Functional loss of TGF-βRII is frequently reported in colorectal cancer (CRC), including bi-allelic mutations in >80% of microsatellite instability-high (MSI-High) [20,21] and roughly 15% of microsatellite stable (MSS) CRCs [22]. TGF-βRII loss is also frequently reported in tumors of

biliary, gastric, brain, and lung tissues [23]. SMAD4 is the most common SMAD family protein disrupted in cancers, and its functional loss or repression has been reported at high frequencies in pancreatic cancer, head and neck squamous cell carcinoma (HNSCC), and CRC, as well as in biliary, bladder, breast, liver, lung, and esophageal cancers [24]. Though point mutations and genetic loss of TGF-β family genes exist with variable frequencies in different cancers, epigenetics also appear to play a significant role in the dysregulation of TGF-β pathway components in cancer. For example, silencing of the TGF-βRII and TGF-βRI genes through hypermethylation has been reported in human mammary carcinomas, and SMAD4 promoter methylation has been reported in advanced prostate cancers [25,26]. Similarly, functional loss of TGF-β family signaling can occur through up-regulation of the I-SMADs (particularly SMAD7) [7,27], increased ubiquitination of the SMAD proteins by SMURF1/2 [4], or increased cytosolic attenuation of SMAD activity by the Ras/Raf/Extracellular signal-regulated kinase (ERK) pathway [25].

Inherited mutations in TGF-β pathway components have also been associated with heredity cancer syndromes. Most notably is juvenile polyposis syndrome (JPS), which is characterized by the development of juvenile polyps of the stomach, small bowel, and large bowel, and increased risk of cancers of the gastrointestinal tract. JPS patients with inherited SMAD4 mutations develop a more severe gastric phenotype and have a worse prognosis compared to those with inherited mutations in BMPR1A [28]. Additionally, germline mutations in TGF-βRs have been associated with increased risk of colon, breast, and ovarian cancers [29–31].

Interestingly, the TGF-β pathway appears to have a duality of function between tumor prevention and tumor promotion [32–35]. In benign epithelia and early-stage tumors, TGF-β is a potent inducer of growth arrest and apoptosis. This is corroborated by the fact that loss of TGF-β family components is often associated with the development of malignant tumors in multiple tissue types. This association has been validated in multiple in vivo mouse models that demonstrate clearly that the loss of TGF-β family signaling elements leads to increased rates of tumor formation in multiple tissues, including the pancreas, stomach, liver, skin, and colon [36–46]. On the other hand, in advanced tumors, TGF-β signaling appears to promote tumor growth, progression, and metastasis, likely reflecting the severe dysregulation at TGF-β family signaling elements [36–39]. The mechanism behind this functional switch from tumor suppressor to tumor promoter remains incompletely understood, but may be related to relative contributions of the canonical and non-canonical TGF-β signaling pathways, differences in intracellular coactivators and corepressors that alter SMAD complex DNA binding activity, or alterations in the tumor microenvironment [33]. This functional switch from tumor suppressor to tumor promoter is known as the TGF-β paradox and is comprehensively reviewed by Principe and colleagues [33].

## 3. TGF-β in Immune Cell Regulation

Importantly, TGF-β ligand remains in the extracellular matrix (ECM) of carcinomas, regardless of the cancer cell's intrinsic ability to respond to TGF-β signaling. In fact, multiple studies have suggested that stromal TGF-β ligand levels are higher in the ECM of tumors with defective TGF-β signaling [47–50]. Thus, even in tumors with the inability to respond to TGF-β, abundant TGF-β ligand remains in the ECM to impinge upon the behavior of adjacent cell populations, including immune cells. The impact of TGF-β signaling on the immune system is significant and well-documented.

It was demonstrated in early murine studies that TGF-β plays a central role in immunomodulation [51]. In the global absence of TGF-β1 expression, mice develop multifocal autoimmune disease, acute wasting, and early death [41,42]. Subsequent studies demonstrated that T cell-specific attenuation of TGF-β signaling also results in autoimmune disease and spontaneous effector T cell differentiation [52]. We now know that TGF-β functionally regulates differentiation of effector and helper T cell sub-populations, inhibiting Th1 and Th2 T cell differentiation while promoting regulatory T cell (Treg) differentiation and suppressing cytotoxic T cell (CTL) activity [53–55]. Importantly, Tregs have a known immuno-inhibitory function and themselves secrete high levels of TGF-β ligand, further perpetuating TGF-β's negative regulation of effector T-cells. Additionally, it has been demonstrated

that inhibition of TGF-β signaling results in increased tumor cytotoxicity and clearance in vivo, owing in part to the enhanced effector functions of CTLs [56,57].

Like the immunomodulatory effects of TGF-β on T cells, the TGF-β immuno-inhibitory role is furthered through its impact on other leukocyte subsets, including natural killer (NK) cells [58], neutrophils [59,60], and macrophages [61]. It has been demonstrated that TGF-β inhibits metabolic activity and interferon-responsiveness of NK cells (via repression of the mTOR pathway) [58]. Perhaps not surprisingly, it has been additionally demonstrated that inhibiting the TGF-β receptor enhances the cytotoxic ability of NK cells in the context of adoptive cell transfer in pre-clinical models [62]. TGF-β has also been implicated in polarization of neutrophils [63] and macrophages [64], particularly in the tumor microenvironment. TGF-β blockade also increases influx of tumor-associated neutrophils with increased cytotoxic/anti-tumor activities whereas, conversely, TGF-β ligand within the tumor microenvironment induces a population of neutrophils with a pro-tumor phenotype [63]. Similarly, TGF-β induces a pro-tumor phenotype in macrophages characterized by up-regulation of anti-inflammation cytokine IL-10 and down-regulation of pro-inflammatory cytokines TNF-$\alpha$ and IL-12 [64].

Taken together, TGF-β is a major negative modulator of the immune system. This suggests a potentially parallel immunosuppressive role in other cell types, including epithelial cells. Additionally, it is perhaps highly relevant that carcinomas with abrogated TGF-β signaling seem to have increased levels of TGF-β ligand in their tumor-associated stroma [47–49]. Elevated TGF-β ligand, while having limited epithelial cell-specific effects in the context of TGF-β desensitization, can impinge on the surrounding immune microenvironment to suppress cytotoxicity and promote immune-tolerance. This may be a major mechanism of immuno-evasion of epithelial tumors with defective TGF-β signaling.

## 4. TGF-β in Epithelial Homeostasis

TGF-β's most well-established role in the epithelial compartment relates to its direct anti-proliferative effects. TGF-β signaling is well known to induce epithelial cell growth arrest through several mechanisms, including direct control over various cyclin-dependent kinase inhibitors, as well as promoting apoptosis and cellular differentiation [2,4,5,33,44,65–67]. While epithelial cell-intrinsic growth control by TGF-β is relatively well characterized, the epithelial cell-intrinsic immunomodulatory control on the surrounding microenvironment by TGF-β and how such modulation may impinge on tumorigenesis or tumor progression is less well understood.

### 4.1. The Immunomodulatory Role of TGF-β in Epithelial Cells and Epithelial Cancers

Several studies have pointed to an immunomodulatory role for TGF-β signaling within the epithelial compartment. For instance, cultured colon epithelial cells continuously exposed to TGF-β ligand were shown to significantly upregulate 15-hydroxyprostaglandin dehydrogenase (PGDH), a protein known to metabolize and decrease the levels of pro-inflammatory prostaglandins. Interestingly, normal colon epithelial cells appear to express relatively high levels of 15-PGDH, whereas 15-PDGH is nearly undetectable in CRC samples. This discrepancy has been attributed to the fact that TGF-β family signaling is disrupted in nearly 80% of CRCs, and suggests an anti-inflammatory role for TGF-β family signaling in colon epithelium [22,68].

Adding to the evidence that TGF-β plays an important immunomodulatory role in colon epithelium are experiments using intestinal epithelium-specific SMAD4 knockout mice [45]. These mice, who have impaired canonical TGF-β signaling within the epithelial compartment, but intact TGF-β family signaling in the surrounding stroma and immune cells, demonstrate increased intestinal epithelial cell expression of genes encoding a variety of pro-inflammatory chemokines and cytokines, including Cxcl5, Ccl20, Ccl8, Il34, and Il18, and this upregulated pro-inflammatory response appears to be at least partially cell-autonomous. Mice with loss of Smad4 expression within their colon epithelial cells were also observed to have an overall 4-fold increase in CD45[+] leukocyte infiltration into the surrounding colonic stroma [45]. Additionally, mouse and human Smad4/SMAD4-deficient intestinal tumors have been associated with increased immune cell accumulation compared to

SMAD4-expressing controls [69–71]. Invasive intestinal tumors of *cis-Apc^+/716^;Smad4^+/-^* mice that exhibited bi-allelic loss of heterozygosity were observed to have marked increased expression of CCL9 and resultant accumulation of immature myeloid cells compared to tumors arising from *Apc^+/716^;Smad4^+/+^* controls [71]. Interestingly, in human CRC samples, CCL15 (the human orthologue to murine CCL9) expression appears to be inversely correlated with SMAD4 expression, and increased tumor CCL15 expression is associated with a three-fold increase in CCR1+ immune cell infiltration [69]. Low SMAD4 expression in human CRC tumors has similarly been associated with increased CD11b+ myeloid cell infiltration [36,55]. Interestingly, a recently published retrospective analysis of human colorectal tumors demonstrated that loss of SMAD4 expression was associated with lower tumor infiltration lymphocytes and a trend towards decreased peritumoral lymphocyte aggregates [72]. These experiments collectively suggest that canonical TGF-β pathway signaling within intestinal epithelial cells and intestinal carcinoma cells has an important role in modulation of surrounding immune cells.

Altered immune cell recruitment due to abrogated TGF-β pathway signaling has additionally been demonstrated in models of HNSCC. In a murine model with epithelial-specific deletion of Smad4 within the oral mucosa, numerous infiltrating leukocytes (including macrophages, granulocytes, and T cells) were observed in the sub-epithelial stroma of *Smad4^−/−^* mucosa compared to controls with *Smad4^+/+^* mucosa. Additionally, *Smad4^−/−^* mucosa had markedly increased expression of several cytokines, including MCP-1, Cxcr7, Csf3, and Ppdp. Of note, mice with *Smad4^−/−^* mucosa spontaneously developed invasive oral tumors whereas *Smad4^+/+^* and *Smad4^+/−^* controls did not [73]. In a parallel experiment, investigators deleted *TGF-βRII* from the head-and-neck epithelium of *Kras* mutant mice and found a significant increase in leukocyte infiltration in the buccal mucosa and HNSCCs of mice with *TGF-βRII^−/−^* mucosa compared to control mice. In this case, leukocytic infiltrate had a predominance of macrophages and granulocytes [49].

Similar immunomodulatory effects of TGF-β pathway signaling have been observed in human mammary cells and in models of mammary carcinoma. In established mammary epithelial cell lines, TGF-β1 suppressed basal and OSM-induced *Cxcl1, Cxcl5,* and *Ccl20* expression [74]. In mouse models of mammary carcinoma, carcinoma-specific deletion of TGF-βRII resulted in increased Gr-1+CD11b+ myeloid cell recruitment to the tumor invasion front, and such recruitment was attributed to upregulation of two chemokine axes: Cxcl5/Cxcr2 and Cxcl12/Cxcr4 [50].

These data, together, suggest an important immunomodulatory role for TGF-β family signaling within epithelial cells. Dysregulation of TGF-β signaling, frequently occurring in pre-malignant and malignant lesions of the gastrointestinal tract, appears to have a substantial impact on the immune microenvironment that may in turn impact tumorigenesis and tumor progression through altered immune cell recruitment (Figure 1). In several of the above-discussed experiments, tumor progression and metastasis was directly attributed to myeloid cell recruitment to TGF-β signaling-deficient tumors due to myeloid production of Matrix Matelloproteinase (MMPs) [50,69–71]. This suggests a novel mechanism of TGF-β's tumor suppressor role in epithelial tissues beyond the well-characterized effects on cell cycle control, although the full impact of immunomodulation by epithelial TGF-β signaling remains incompletely understood.

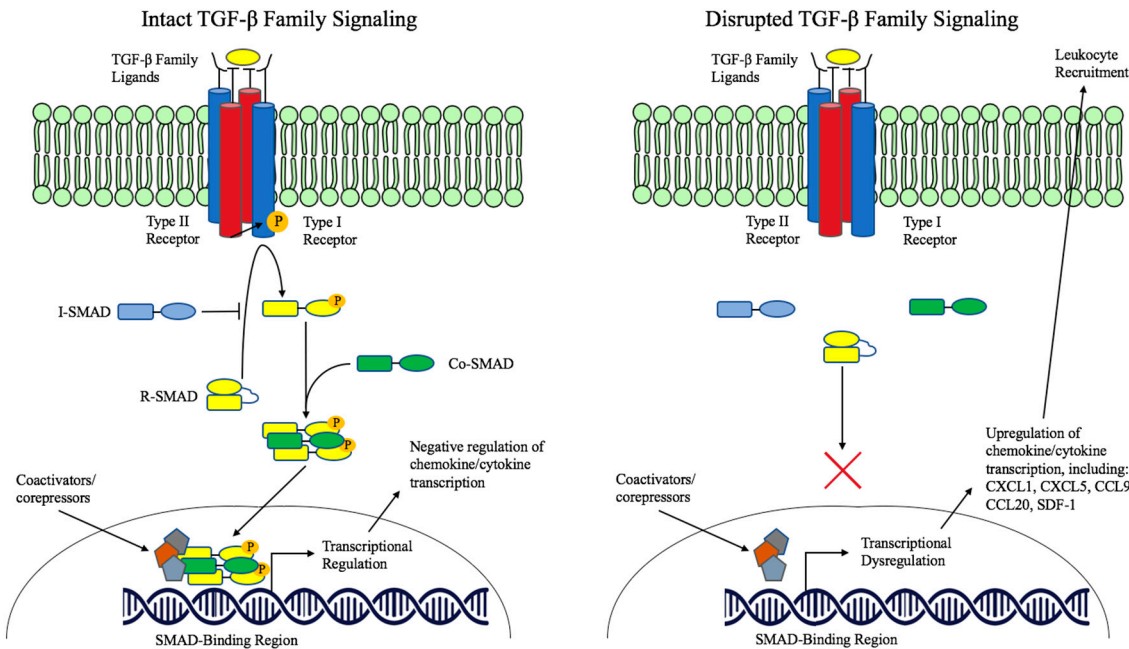

**Figure 1.** Immunomodulatory role of TGF-β family signaling in epithelium.

### 4.2. TGF-β Dysregulation in Inflammatory Bowel Disease

A careful balance of pro- and anti-inflammatory signals in the intestinal epithelium is critical maintaining intestinal homeostasis. The intestine is home to thousands of microbial species [75], and the intestinal mucosa must extinguish invading pathogens quickly to prevent organismal infection due to minor mucosal injuries. At the same time, the inflammatory response to resident bacteria must be tempered and self-limited to prevent pathologic intestinal inflammation. Dysregulation of this equilibrium between pro- and anti-inflammatory signals in the intestine is thought to be a major contributing factor to inflammatory bowel disease (IBD), and the dysregulated pathways that contribute to the development of IBD is an active area of research [76].

TGF-β pathway signaling may play a critical role in extinguishing pro-inflammatory signals in response to resident microbes in the intestine. In an intestine-specific dominant-negative *TGF-βRII* (dnR2) transgenic mouse model, dnR2 mice were healthy when housed under specific pathogen-free conditions but quickly developed spontaneous colitis, weight loss, severe diarrhea, and hematochezia when housed in normal rodent housing in the presence of standard microbes. The intestinal mucosa of dnR2 mice was found to have significantly increased expression of *Il-2*, *Il1-β*, *IFN-γ*, *IL-10*, and *TGF-β1*, and dnR2 mice appeared to be highly susceptible to dextran sulfate sodium (DSS)-induced colitis compared to wild type mice [48].

Interestingly, inhibitory-SMAD (SMAD7) protein levels have been found to be increased in mucosal biopsy samples of patients with Crohn's disease when compared to healthy controls [27,77]. Accordingly, SMAD3 phosphorylation levels, a marker of canonical TGF-β pathway activity, was markedly reduced in mucosal samples of Crohn's patients compared to mucosal samples from healthy controls [27]. Importantly, it was demonstrated that *Smad7* antisense therapy reduced SMAD7 protein levels, increased levels of phosphorylated SMAD3, and decreased levels of mucosal pro-inflammatory cytokines including TNF-α and IFN-γ [27]. Phase 1 clinical trials of oral Smad7 knockdown therapy demonstrated clinical safety [78] and a double-blind phase 2 trial found that patients with Crohn's disease who received Smad7 knockdown therapy had significantly higher rates of remission and clinical response than those who received the placebo [79].

## 5. Conclusions and Unanswered Questions

While TGF-β's roles in modulating epithelial cell proliferation and immune cell activation have been well characterized, the role of TGF-β signaling within epithelial cells as it impinges on immunomodulation is less well understood. Several murine experiments have recently drawn attention to the immunomodulatory role of TGF-β family signaling in epithelial cells and epithelial cancers [22,27,45,47–50,68–71,73,74,77–79]. It appears that canonical TGF-β signaling within epithelial cells plays a role in suppressing pro-inflammatory chemokine and cytokine expression, and that loss of functional TGF-β signaling results in up-regulation of multiple pro-inflammatory chemokines and cytokines, resulting in altered immune cell recruitment. Though in some contexts this altered leukocyte recruitment may directly impinge on epithelial cancer progression, such as through increased immature myeloid recruitment and subsequent MMP secretion, exactly how this altered chemokine/cytokine expression profile impinges on the immune system and its implications for tumorigenesis and tumor progression remains largely unexplored. Furthermore, whether the altered landscape of chemokine/cytokine production that occurs because of aberrant epithelial TGF-β signaling has implications for leukocyte activation, differentiation, or behavior in the epithelial microenvironment remains unknown.

Furthering the intricacy of this scenario of altered immune cell recruitment towards TGF-β signaling-deficient epithelium and epithelial cancers is the observation that tumors with altered TGF-β signaling appear to have increased TGF-β ligand in their tumor-associated stroma [47–50]. While increased TGF-β ligand abundance is generally felt to be an important mediator of immune-evasion in tumors with defective TGF-β signaling, how altered epithelial cell chemokine/cytokine expression in this context may further impinge on leukocyte recruitment, differentiation, cytotoxicity, and behavior beyond the known immunomodulatory-effects of TGF-β ligand on leukocytes is largely unknown.

Developing a more sophisticated understanding of the immunomodulatory role of TGF-β family signaling within epithelial cells has the potential to greatly improve our understanding of TGF-β's tumor suppressive role beyond its well-known anti-proliferative effects. Additionally, such investigation may allow us to understand how the loss of functional TGF-β signaling in epithelial tumors, a relatively frequent event, may lead to targetable alterations in the immune microenvironment. Such insight could have therapeutic implication for IBD patients and for patients with TGF-β-deficient epithelial tumors.

**Author Contributions:** P.M.S. did background research, drafted the manuscript, created the figure, and participated in editing the final manuscript and making reviewer-suggested revisions. A.L.M. and R.D.B. both provided oversight for the background research, manuscript drafting, and figure creation and also each provided edits prior to manuscript submission as well as in response to reviewer suggestions.

**Funding:** This research was funded by the National Cancer Institute, grant number 1F32CA236309-01.

**Conflicts of Interest:** The authors declare no conflict of interest.

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
