# Peer review of "Immunomodulatory Effects of TGF-β Family Signaling within Intestinal Epithelial Cells and Carcinomas"

_gastrointestdisord, doi:10.3390/gidisord1020024_

Round 1

Reviewer 1 Report

A very nice review by Smith, et al.   

Please expand on the non-SMAD signlaing molecular that act as the effector molecules for non-SMAD activity. This is especially in the context of metastatic disease.  This is not fully described in the literature but worthy of a bit more analysis in this review article.

Please get into a little more detail regarding the role of TGF-beta in inducing changes to immune cells- activity of NK cells, T cells, etc. as well as polarization of macrophages.  This is a critically important TME context.

Reviewer 2 Report

Dear Editor,

Gastrointetinal disorders

 The authors discribed detailed TGF-b sinlaling in leukocytes and intestinal epithelial cells. This review is well written, and especially Smad7 knockdown therapy for Crohn’s disease is very immpressive. I think this paper is good for publication in the present form.

Overall recommendation: 

 Good for publish in the present form.
